# RoboFlow: a Data-centric Workflow Management System for Developing AI-enhanced Robots

**Qinjie Lin**[*], **Guo Ye**[*], **Jiayi Wang, Han Liu**
Department of Computer Science, Northwestern University

**Abstract:** We propose RoboFlow, a cloud-based workflow management system orchestrating the pipelines of developing AI-enhanced robots. Unlike most traditional robotic development processes that are essentially process-centric, RoboFlow is data-centric. This striking property makes it especially suitable for developing AI-enhanced robots in which data play a central role. More specifically, RoboFlow models the whole robotic development process into 4 building modules (1. data processing, 2. algorithmic development, 3. back testing and 4. application adaptation) interacting with a centralized data engine. All these building modules are containerized and orchestrated under a unified interfacing framework. Such an architectural design greatly increases the maintainability and re-usability of all the building modules and enables us to develop them in a fully parallel fashion. To demonstrate the efficacy of the developed system, we exploit it to develop two prototype systems named "Egomobility" and "Egoplan". Egomobility provides general-purpose navigation functionalities for a wide variety of mobile robots and Egoplan solves path planning problems in high dimensional continuous state and action spaces for robot arms. Our result shows that RoboFlow can significantly streamline the whole development lifecycle and the same workflow is applicable to numerous intelligent robotic applications[2].

**Keywords:** AI-enhanced robots, robotic development workflow management, data-centric development, cloud-based robotic development

## 1   Introduction

We propose RoboFlow, a cloud-based workflow management system for developing data-centric and AI-enhanced robots. This work is done in the context that significant progresses have been made in robotics development and a paradigm shift from process-centric development to data-centric development is being witnessed, especially for learning robots. Specifically, traditional robot development workflow [1, 2, 3, 4, 5, 6] is essentially process-centric, which emphasizes more on designing, developing and integrating different "processing modules" interacting and inter-operating with each other in a complex fashion. Such a process-centric robotic development model, though natural for humans, is not suitable for developing modern AI-enhanced robotic systems (aka., learning robots) that are essentially data-centric [7, 8, 9, 10, 11, 12]. Some key reasons are that the development processes of AI-enhanced robots generally involve managing and interacting with massive amounts of data, and even after the systems have been deployed, continuous modification and improvement are still needed when more data get acquired. Such an extra "data-centric" dimension of learning robots causes a dramatic increase in both coding complexity and maintaining complexity of the traditional process-centric robotic development workflow, thus a new data-centric robotic development workflow is crucially is needed. To bridge this gap, we propose RoboFlow.

A high-level overview of RoboFlow is illustrated in Figure 1. In the most abstract fashion, the RoboFlow system divides the whole pipeline of developing AI-enhanced robots into 4 building modules (1. data processing, 2. algorithmic development, 3. back testing and 4. application adaptation) interacting with a centralized data engine. Specifically, the data engine can be viewed as an "oracle" that abstracts out all the data management details and is interacting (e.g., being queried or manipulated) with all the 4 building modules in an asynchronous fashion. Centered around the

---

[*]Authors contributed equally
[2]Project site: https://sites.google.com/u.northwestern.edu/roboflow

Blue Sky Papers, 5th Conference on Robot Learning (CoRL 2021), London, UK.

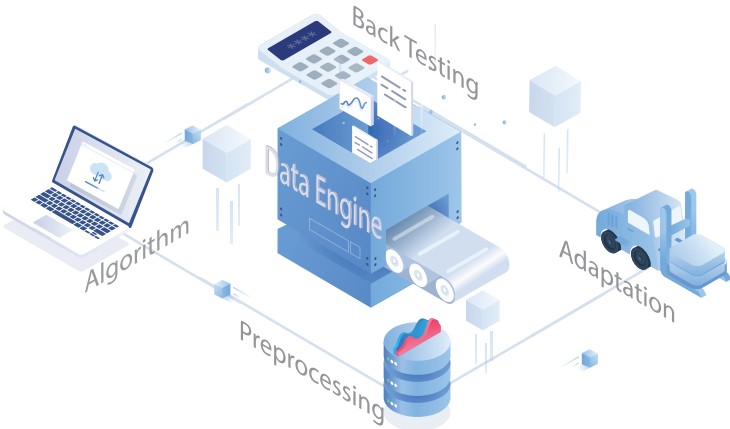

Figure 1: An overview of RoboFlow. The robot development pipeline consists of 4 modules interacting with a centralized data engine. The data engine manages the large-scale dataset, and publishes data from robot. The data preprocess module encodes raw input to data that algorithm development module can easily parse. The algorithms development module develops customized control policy. The back testing module tests the policy in various environments. The application adaption module deploys the learned policy in real world.

data engine, the 4 building modules follow an iterative spiral model. Unlike the classical spiral model[13] for software development, these building blocks mainly interact with data engine, thus can be developed in a fully parallel fashion and each module could have different "versions". Such a data-centric design dramatically decreases the developing and maintaining complexities, making it especially suitable for developing AI-enhanced robots. The framework in Figure 1 is quite generic and described in a fully abstract way. To understand this framework better, we put it in a concrete context of developing learning robots. From a learning perspective, the data processing module transforms and encodes data into a state that other modules can easily parse. The algorithm module represents the planning policy of the robots, which is in charge of the robots' actions. The back testing module evaluates the policy of the learning robot based on large-scale collected or simulated data. The application adaptation module puts the developed policy into real-world environments once the back testing meets a designed criterion. More details about these modules will be provided in Section 3.

This paper has 3 major contributions: (i) we proposed a novel data-centric robot development model which improves upon the traditional routines in terms of development flexibility and maintainability. (ii) We implement a first prototype system using containerization techniques (specifically, Docker[14] and Kubernetes [15]). (iii) Using this system, we develop two platforms: Egomobility and Egoplan. Egomobility exploits deep reinforcement learning to provide navigation ability for turtlebot and Egoplan uses imitation learning to solve complex motion planning problems for sawyer robots. These studies demonstrate the efficacy of RoboFlow in various intelligent robotics applications. The rest of this paper is organized as below. Section 2 introduces related work. Section 3 describes implementation of RoboFlow. Section 4 presents two platforms to showcase the usefulness of RoboFlow.

## 2 Related Work

The two most relevant lines of work related to RoboFlow are the workflow management system and data management engine for data-centric robot systems. For the workflow management system, earlier robot development pipelines try to simplify the workflow of robotics development and reduce re-programming efforts. For example, Fetch robotics[16] launches a Workflow Builder which allows customers to design, implement, and redesign their own workflows. But their available tools are not designed for large-scale dataset processing. In the machine learning field, frameworks like Kubeflow[17] and MLFlow[18] have been developed to manage the workflow of model development but they are not designed for robot development. For the data management engine, researchers in bioinformatics community developed system tools to manage data workflows in an end-to-end fashion[19, 20]. Nevertheless, in robotics, few works emphasize on the data-centric aspect of the development processes except some proposals related to cyber-physical systems which view robots as data-gathering nodes [21, 22, 23, 24, 25, 26, 21]. In recent work, Farzad et al. [27] proposed a cloud framework aiming to facilitate the development of IoT applications. but it is not straightforward on how to apply it to the more heavy-weighted data-centric robotic systems.

# 3  System Architecture and Software Implementation

The RoboFlow architecture builds upon the containerization and container orchestration techniques. More specifically, a container platform (e.g., the Docker) packages applications so that they can access a specific set of resources on a physical or virtual host. The main benefit, especially for developers, is that containers isolate different applications and are elastic, i.e., come and go as demanded by need. This is particularly useful for developing massive robotic systems in which developers may contribute code in different programming languages and application frameworks. In such scenarios, we could exploit a container platform to establish many containers to isolate and manage all the developed applications. To manage containers at scale, we can utilize a container orchestrate system (e.g., Kubernetes, Docker Swarm) to automate the deployment, management, scaling, networking, and availability of all the containers. In the rest of this section, we describe the system architecture and software implementation of the RoboFlow system.

**System Architecture.** Figure 2 illustrates the system architecture of RoboFlow. It has 4 essential modules(1.Data Preprocess 2.Algorithm Development 3.Back Testing 4.Application Adaptation) interacting with a centralized data engine. Each module is employed into a containerized environment by bundling it together with all related configuration files, libraries and dependencies. These module containers run isolated processes on the system, thus enabling RoboFlow to be developed in a fully parallel fashion. In addition, any change in these module containers is recorded, making version control easy to implement. These containers exploit DDS (Data Distribution Service) to manage real-time communication between them and the data engine. DDS implements a publish-subscribe pattern for sending and receiving data and each process running in RoboFlow are considered as DDS nodes to communicate data with other process. Also, a networked filesystem named Glusterfs is also ultilized for sharing large-scale files (e.g., large neural network models or training datasets) between modules. Such integration of DDS and Glusterfs make RoboFlow suitable for developing data-driven methods on robotics. RoboFlow also provides a web-based frontend to ease developers to monitor, analyze, and manage the robotic development process.

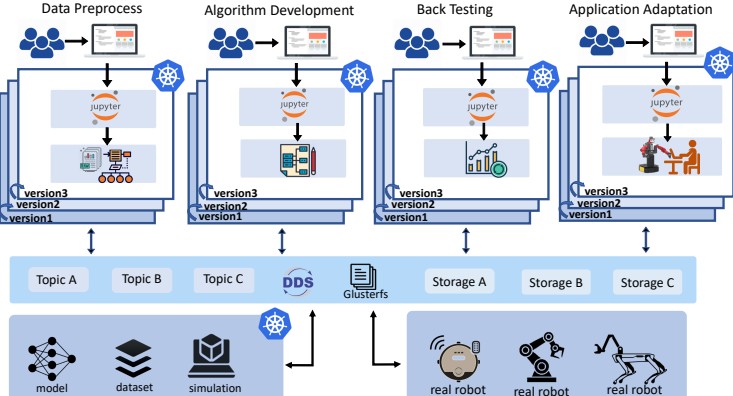

Figure 2: The architecture of RoboFlow. RoboFlow provides a graphical user interface for developers to access all containerized modules. These modules interact with the data engine through shared storage and DDS topics. The data engine consists of large-scale datasets and model data stored in the shared folder, simulation containers and robots connected to the RoboFlow system through DDS topics.

**Software Implementation.** To implement RoboFlow, we wrap each module as a `docker` image. For this, we specify the software environment (e.g., the operating system distribution and pre-installed packages) of an image in a `Dockerfile`. The obtained module images are installed with `juypter lab` and ROS2. Some modules may need additional packages. For example, the algorithm development module is equipped with the deep learning libraries `tensorflow` and `pytorch`, while the data engine image utilizes several robot simulators like `Gazebo`, `Stage` and `OpenRave`.The successfully built images are stored in a cloud-based storage space named `cargo`, which can be viewed as on-premise `dockerhub` of the RoboFlow system. During the run time, each instance of a built image is deployed as a container and RoboFlow utilizes `Kubernetes` to deploy and manage these containers. We exploit the `React` javascript library to implement a graphical user interface (GUI). Through this GUI users can choose desired versions of the docker images and allocate computational resources (e.g., CPUs or GPUs) to the container being created. Once such configuration information is submitted, it is turned into a `YAML` file which will be serialized and deployed by `Kubernetes`.

# 4 Two Case Studies and Performance Evalaution

In this section, we exploit RoboFlow to develop two prototype systems named "Egomobility" and "Egoplan". Egomobility provides a general-purpose navigation platform for managing a wide variety of mobile robots and Egoplan is a motion planning platform for robot arms. To demonstrate the efficacy of RoboFlow, we also conduct some performance analysis of the two case studies.

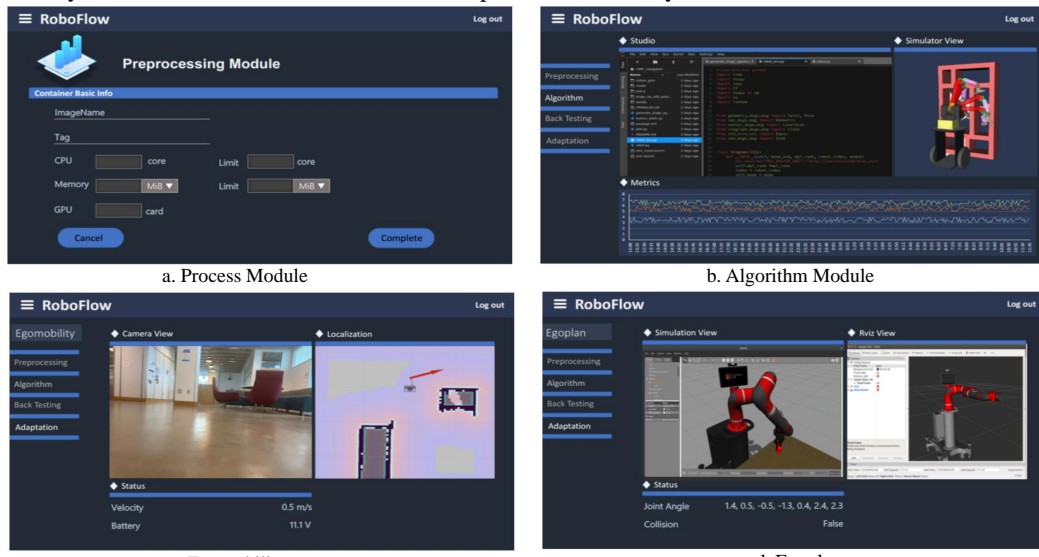

a. Process Module   b. Algorithm Module
c. Egomobility   d. Egoplan

Figure 3: Graphical User interface of RoboFlow. Figure a illustrates the edit page of preprocess module, which allows developers to specify the image name, version, required CPU number, memory size, and GPU number. Figure b illustrates the algorithm module, enabling users to access the code editor and simulator view, monitor the progress of a learning robot training process. Figure c illustrates the Egomobility platform. This adaptation page provides real-time camera stream and current position, velocity and battery usage. Figure d illustrate the Egoplan, which provides the simulation views, rviz view and status panel.

**Case Study 1: The Egomobility platform for Mobile Robots.** In this study, we exploit RoboFlow to develop a mobile robot navigation platform named Egomobility, which is a data-centric AI-enhanced robot system using data from a `Stage` simulator [28] as training data. Within this environment, a mobile robot takes 3 raw laser frames and its velocity as input. Our goal is to train a reinforcement learning policy that outputs a velocity guiding the robot avoiding dynamic obstacles. The deployment page of Egomobility is provided in Figure 3.

**Case Study 2: The Egoplan Platform for Arm Robots.** In this study, we exploit RoboFlow to develop a robot arm motion planning Platform named EgoPlan. More specifically, we exploit the NEXT (Neural Exploration-Exploitation Trees) algorithm proposed in[29] to learn a motion plan policy for solving path planning problems in 7-dimensional state space and 7-dimensional action space. In each planning task, we simulate a robot arm and a shelf in the `openrave` [30] simulator in our data engine. Each level of the shelf is horizontally divided into multiple bins. The task for the robot arm is to plan a collision-free path from an initial pose to grab an object placed in a shelf. The main strategy is to exploit BIT* [31] to solve the planning problem in a brutal force way but collect the data to learn a smarter motion planning policy.

**Performance.** Through the above two case studies, the benefit of RoboFlow for managing a data-centric robotic system development pipeline is quite obvious: RoboFlow enables developers to develop all the component modules in a fully-parallel fashion. For example, in developing Egomobility, algorithm developers can join in the RoboFlow to develop the navigation algorithm independently without interfering with each other. The developers for the application adaptation module can simultaneously test many learned policies without the need of complex communication processes. In addition, due to the usage of sophisticated container-orchestration techniques, the resulting systems developed by RoboFlow is much more reliable and maintainable.

**Conclusion and Discussion.** We propose RoboFlow, a data-centric workflow management system orchestrating the pipelines of developing AI-enhanced robots. The data-centric features of RoboFlow illustrate high maintainability and re-usability of each module. We hope this data-centric model brings a new approach to the community.

**Acknowledgments**

Han Liu's research is supported by the NSF BIGDATA 1840866, NSF CAREER 1841569, NSF TRIPODS 1740735, DARPA-PA-18-02-09-QED-RML-FP-003, along with an Alfred P Sloan Fellowship and a PECASE award.

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
