# OpenReview forum: "RoboFlow: a Data-centric Workflow Management System for Developing AI-enhanced Robots"
_robot-learning.org/CoRL/2021/Conference/Blue_Sky — CoRL 2021, Blue Sky_

### Official Review · Reviewer_VRc6 · 2021-08-04

**Novelty:** Very Good
**Impact:** 4
**Clarity Of Presentation:** Good

**Recommendation:**

Weak Accept: I recommend accepting the paper, but will not argue for my recommendation if the majority of other reviewers have a different opinion.

**Summary:**

This is a unique software systems paper that describes a hypothetical robot development platform based on virtualization technologies currently under very active use in the systems community.  Two simple case studies implementing RL and motion planning algorithms are demonstrated.

**Summary Of Recommendation:**

The paper is woefully light on details, but has good potential to spark discussion about future pipelines for data-driven robot development. In my opinion, the field needs to revise its outlook on middleware as being "just use ROS" which does a terrible job of integrating modern learning workflows, and this paper is a step in the right direction. The paper itself could benefit from describing more clearly their vision for a data-driven development framework, what a developer's workflow would look like, how such a system could aid in managing datasets, domain transfer, etc.

---

### Official Review · Reviewer_ShNX · 2021-08-29

**Novelty:** Fair
**Impact:** 2
**Clarity Of Presentation:** Good

**Recommendation:**

Strong Reject: I recommend rejecting the paper and will argue for my recommendation even if other reviewers hold a different opinion.

**Summary:**

The article proposes RoboFlow, a cloud-based workflow management system that divides the robot software pipeline into four modules: data processing, algorithmic development, back testing and application adaptation.  Each of those modules interact with a centralized data engine with manages access to one or more datasets.  This architecture is motivated by the increased use of “data-centric” algorithms that are trained on large quantities of data.  The authors present a prototype variant of RoboFlow and demonstrate it in two domains.

This paper would be more suitable as a traditional conference or journal article than a Blue Sky submission.  Although the paper is forward thinking (it does propose a long term challenge and future architecture), so do many conference papers as well.  The main issue here is that there is not enough space in 4 pages to fully describe the proposed system and support the claims made by the authors.  Specifically:

* The claim that existing architectures do not support parallel development is unfounded and inaccurate.  I strongly encourage the authors to revisit and rephrase.

* It is not clear what actual gains would be achieved from the proposed system. Is it computational efficiency?  Reduced data bandwidth?  Lower memory costs?

* To address the above, the system should be benchmarked against similar systems and statistics should be reported

* Due to page limitations, the details of both the architecture and the performed case studies are insufficient

The work would greatly benefit from being presented in a longer format at a different venue.

**Summary Of Recommendation:**

This paper would be more suitable as a traditional conference or journal article than a Blue Sky submission.  Although the paper is forward thinking (it does propose a long term challenge and future architecture), so do many conference papers as well.  The main issue here is that there is not enough space in 4 pages to fully describe the proposed software architecture and support the claims made by the authors.

---

### Decision · Program_Chairs · 2021-10-01

**Decision:**

Accept

**Comment:**

The paper proposes a new way of looking at the robot pipeline -- from the module oriented to the data-centric point of view. While the paper is sketch of an idea, it provides a novel point of view, likely to spark a good discussion and potentially inspire different way of looking at the robotics stack.